# Introducing the trier univalence neutrality ambivalence (TUNA) database: A picture database differentiating complex attitudes

Lena Hahn[1,2]*, Benjamin Buttlar[1], Ria Künne[1], Eva Walther[1]

**1** Department of Psychology, Trier University, Trier, Germany, **2** Leibniz Institute for Psychology (ZPID), Trier, Germany

☯ These authors contributed equally to this work.
* hahnl@uni-trier.de

**Data Availability Statement:** Supplemental material including data, analysis script, app, original sized pictures, unformatted pictures, screenshots as well as the validation table is

## Abstract

Using validated stimulus material is crucial for ensuring research comparability and replicability. However, many databases rely solely on bidimensional valence ratings, ranging from negative to positive. While this material might be appropriate for certain studies, it does not reflect the complexity of attitudes and therefore might hamper the unambiguous interpretation of some study results. In fact, most databases cannot differentiate between neutral (i.e., neither positive nor negative) and ambivalent (i.e., simultaneously positive and negative) attitudes. Consequently, even presumably univalent (only positive or negative) stimuli cannot be clearly distinguished from ambivalent ones when selected via bipolar rating scales. In the present research, we introduce the Trier Univalence Neutrality Ambivalence (TUNA) database, a database containing 304,262 validation ratings from heterogeneous samples of 3,232 participants and at least 20 ($M = 27.3$, $SD = 4.84$) ratings per self-report scale per picture for a variety of attitude objects on split semantic differential scales. As these scales measure positive and negative evaluations independently, the TUNA database allows to distinguish univalence, neutrality, and ambivalence (i.e., potential ambivalence). TUNA also goes beyond previous databases by validating the stimulus materials on affective outcomes such as experiences of conflict (i.e., felt ambivalence), arousal, anger, disgust, and empathy. The TUNA database consists of 796 pictures and is compatible with other popular databases. It sets a focus on food pictures in various forms (e.g., raw vs. cooked, non-processed vs. highly processed), but includes pictures of other objects that are typically used in research to study univalent (e.g., flowers) and ambivalent (e.g., money, cars) attitudes for comparison. Furthermore, to facilitate the stimulus selection the TUNA database has an accompanying desktop app that allows easy stimulus selection via a multitude of filter options.

## Introduction

The inferences that can be drawn from psychological research depend on the validity of the study materials. Thus, validated stimulus databases have been developed to help researchers

available via the Open Science Framework: https://osf.io/6br4d/.

**Funding:** LH and BB received funding from the Research Fund by Trier University (4,895.25 €). The publication was supported by the Open Access Fund of Universität Trier and by the German Research Foundation (DFG). The funders had no role in study design, data collection and analysis, decision to publish, or preparation of the manuscript.

**Competing interests:** The authors have declared that no competing interests exist.

select appropriate study materials. Validated databases exist for video [1,2], auditory [3], and text stimuli [4–6], while most databases focus on picture stimuli. The various databases that entail picture validation have different foci, for example, on affective [7–10], moral [11], face [12], animal [13], or art pictures [14].

As a standard dimension, most databases capture valence as the evaluation of the stimuli from negative to positive [15]. This bipolar conceptualization suggests that attitudes are univalent positive, meaning they are entirely positive, or univalent negative, meaning they are entirely negative. However, attitudes can also be neutral, indicating neither positivity nor negativity or ambivalent, reflecting both strong positivity and negativity at the same time [16]. These neutral and ambivalent attitudes cannot be differentiated with standard measures using bipolar scales from (very) negative to (very) positive [17,18].

Importantly, univalence, neutrality, and ambivalence have fundamentally different affective, cognitive, and behavioral consequences [16]. Affective consequences of ambivalence are, for example, increased negative emotions like uncertainty, anxiety, or irrationality [19]. Additionally, when people experience ambivalence, they also report higher arousal [20]. Because ambivalence elicits these aversive experiences and is related to negative affect [21], it results in other downstream consequences. Cognitive consequences of ambivalence can be biased information processing (e.g., one-sided information search; [22]) or compensatory perception of order (e.g., higher belief in conspiracies; [19]). People also use behavioral strategies to cope with the experience of ambivalence. That is, people who experience ambivalence procrastinate and postpone their decision regarding an ambivalent attitude object [23]. In contrast, neutrality is not related to negative affect and does not instigate those coping strategies, rather it is characterized by a lack of action [17]. We, therefore, argue that ambivalence and neutrality do not only differ conceptually but affect people's responses to stimulus materials. Indeed, previous research indicated differences in the response times and response patterns regarding neutral and ambivalent stimuli [24]. By selecting stimulus material based on valence ratings that cannot account for these differences, researchers may thus draw invalid inferences from their (psychological) studies.

Invalid inferences may particularly occur if stimulus material is selected based on presumably similar attitudes measured via bipolar valence scales. In fact, when selecting validated stimuli via commonly used bipolar valence ratings, researchers may inadvertently use ambivalent, instead of neutral stimuli. This was demonstrated by Schneider et al. [20] for the International Affective Picture System (IAPS; [8])—one of the most popular picture databases. They showed that pictures that received ratings in the middle of a bipolar valence scale, were related to ambivalence instead of neutrality. This was especially the case when the arousal ratings were high. Thus, indicating that the midpoint of a bipolar ambivalence scale cannot be interpreted unambiguously.

Bipolar scales may not only bias the interpretation of ratings on their midpoint, but they can also lead to incorrect inferences about ratings at the endpoints of bipolar valence scales. Using bipolar scales, Norris et al. [25] demonstrated that it is not possible to differentiate between the evaluation of healthy (e.g., fruits) and unhealthy foods (e.g., sweets) as both were evaluated rather positively. Only by using measures that allow a differentiation of complex attitudes, were they able to uncover the differential attitudes underlying healthy and unhealthy foods, showing that healthy foods are less ambivalent than unhealthy foods. This illustrates that stimulus materials must be validated without relying on bipolar conceptualizations, and instead should be validated and selected via ratings that allow a differentiation between univalence, neutrality, and ambivalence.

## Differentiating univalence, neutrality, and ambivalence

To achieve a clear differentiation between univalence, neutrality, and ambivalence, researchers use two unipolar scales to capture positivity and negativity separately [18]. On these split semantic scales, respondents indicate how positive [negative] an object is regardless of its negative [positive] aspects, with the endpoints of the scale labeled not positive [negative] at all to very positive [negative]. Both ratings are then combined into an index—most commonly the similarity intensity model index (SIM-Index; [26]): (Positivity + Negativity)/2 - |Positivity-Negativity|. This formula incorporates the intensity of the components (i.e., [Positivity + Negativity]/2) corrected by the polarization (i.e., | Positivity–Negativity |). Therefore, univalence is present when the rating on one scale is high and on the other scale low (i.e., high intensity and high polarization). For example, positivity is indicated when participants rate high on the positivity scale and low on the negativity scale. Neutrality is present when the ratings on both scales are low (i.e., low intensity and low polarization), and ambivalence is present when the ratings on both scales are rather high (i.e., high intensity and low polarization). Because these measures represent a relatively stable underlying structure of negative and positive associations this ambivalence is often called structural, objective, or potential ambivalence [20].

In contrast to potential ambivalence, felt ambivalence is the subjective experience of ambivalence. This felt ambivalence encompasses the experienced conflict [27] when negative and positive associations towards an attitude object become accessible simultaneously [28]. Felt ambivalence can be measured following the tripartite model of attitudes [29] by asking people how conflicted (i.e., affect component), mixed (i.e., cognitive component), and indecisive (i.e., behavioral component) they feel [30]. Moreover, a single item, asking the extent to which people experience conflicting thoughts and/or feelings regarding the attitude object, can be used to assess felt ambivalence [30,31]. Because felt ambivalence has mostly been associated with the downstream consequences of ambivalence [23] researchers should consider felt ambivalence along with potential ambivalence when selecting stimuli.

Even though researchers have advocated for incorporating these ambivalence ratings in validation studies [9,11,20], there is still no database including ratings to distinguish between univalence, neutrality, and ambivalence. In fact, only Norris et al. [25] presented potential and felt ambivalence ratings for a subset of images (150 images) from the Food Cast research image database (FRIDa; [32]). Thus, based on the breadth of existing databases, researchers might be prone to drawing incorrect inferences because they cannot account for the full complexity of attitudes.

## Introducing the TUNA database

In the present paper, we, therefore, introduce the Trier Univalence Neutrality Ambivalence (TUNA) picture database: The first database including validation ratings that allow to differentiate univalence, neutrality, and ambivalence. In specific, the TUNA database allows researchers to differentiate univalence, neutrality, and ambivalence by providing validation ratings for positivity and negativity measured independently via split semantic scales. Moreover, the TUNA database provides validation ratings for felt ambivalence as the meta-cognitive experience of conflict. Thereby, the TUNA database presents a comprehensive picture database that surpasses previous databases and enables researchers to select stimuli even if they involve more complex attitudes.

A challenge for the creation of the TUNA database was finding picture stimuli that are ambivalent. Previous ambivalence research predominantly used word stimuli (e.g., questionnaires; [33–35]). This might be due to the fact that ambivalent attitude objects like abortion [36], drug use [34], physical exercise [37], organ donation [38], or romantic partners [33], are

difficult to unambiguously capture in photographs (e.g., does an injection needle signify drug use or a life-saving medication injection). However, some ambivalence research has utilized picture stimuli (e.g., [25,31,39–43]). These studies predominantly focus on attitudes toward food [25,31,39,40,42]. Even when the emphasis is not on food, the picture may still feature food-related elements (e.g., plastic-packed or unpacked food when investigating plastic related ambivalence; [41]) or an explanation of the picture was provided before the evaluation of the pictures [43]. The rationale behind this food-centric focus lies in the distinctiveness of food as a source of particularly salient univalent (e.g., fruits; [25]), neutral (e.g., bread or oats; [44]), and ambivalent (e.g., meat or chocolate; [35,45]) attitudes.

While the majority of images in the TUNA database portray food, it also includes a diverse array of potentially ambivalent attitude objects. For example, in the study exploring ambivalence in the IAPS pictures, various attitude objects such as a hammer, wolf, chair, and mug elicited ambivalence [20]. Consequently, the TUNA database also encompasses pictures of potentially ambivalent items (e.g., hammer, mug, chair) and animals (e.g., dog, cow) as well as other ambivalent content like vehicles or money. Because ambivalence research oftentimes incorporates univalent and/or neutral control categories (e.g., [24,31]), the TUNA database additionally includes pictures of attitude objects that are typically used in research to study univalent (e.g., flowers) and neutral (e.g., toaster) attitudes [7,9,46,47]. Hence, the TUNA database encompasses a variety of univalent, neutral, and ambivalent picture stimuli.

We took several measures to maximize the usability of the TUNA database. First, we aimed to keep the TUNA database compatible with previous databases. Therefore, we oriented our validation on the highly cited food-pics [47] and the FRIDa [32] databases and assessed similar validation ratings including, for example, recognizability, familiarity, desire to eat, palatability, valence, arousal, and complexity. Furthermore and as other databases that include food pictures, the validation tables for the TUNA database include nutritional information for the food images (e.g., [47]) and information about image properties (i.e., color, size, intensity, contrast, complexity; [47]). Second, the TUNA database includes ratings on anger, disgust, and empathy. These affective measures have been shown to be relevant for food consumption [48–50]. Additionally, anger and disgust are correlates of ambivalence toward meat [51,52]. For an analysis of how validation ratings relate, see Buttlar et al. [53]. Third, the database encompasses food pictures in various forms (e.g., raw vs. cooked, non-processed vs. highly processed) because processing has been shown to influence the evaluation of food [48,49]. Fourth, besides the total sample, the validation ratings can be used independent for the representative US and the German student sample. Because research has shown that validation ratings can differ depending on gender [9,47] or diet [47], the validation ratings can also be retrieved for female, male, veg*ans (i.e., vegetarians and vegans) and omnivores separately.

In sum, the TUNA database is a highly versatile picture database. The TUNA database encompasses various validation ratings (i.e., positivity and negativity, which were also combined into a potential ambivalence index, felt ambivalence, valence, arousal, anger, disgust, empathy, palatability, desire to eat, familiarity, recognizability, and complexity), objective image characteristics (i.e., color, size, intensity, within-object contrast, and object complexity) as well as nutritional information for food pictures (i.e., protein, carbs, fat, and kcal per 100g). This extensive range of information allows researchers to consider various facets of human stimulus perception and (emotional) response when selecting their study material. Additionally, the total of 304,262 validation ratings were from a heterogeneous sample of 3,232 participants, and at least 20 ratings per self-report scale per picture were collected. However, to account for sample specifics and enable researchers with a specific target sample, the validation data can be retrieved for the different subsamples (i.e., German student, representative US, male, female, veg*ans, omnivores). Furthermore, to maximize accessibility and streamline the

research process, the TUNA database is accompanied by a user-friendly access tool (https://osf.io/fys36/). This tool facilitates the selection of suitable stimulus material based on the extensive validation ratings. Hence, making the TUNA database an invaluable asset for researchers seeking a comprehensive resource for picture stimuli.

The main aim of this paper is to introduce the TUNA database and demonstrate the need to differentiate between univalence, neutrality and ambivalence. In the following, we, therefore, describe the TUNA database and its validation. Based on the validation ratings, we then exemplify the pitfalls of relying on bipolar valence scales to measure attitudes and the advantages of selecting stimulus material based on split semantic scales. While researchers often try to use neutral stimulus material as a control condition, they typically select the stimuli by asserting their neutrality without consulting validated database or by relying on databases with bipolar valence scales. This approach might lead to false conclusions based on the people's complex attitudes. Therefore, we first tested whether stimulus material that has been suggested to be univalent (i.e., fruits; [25]), neutral (i.e., carbs; [44]), and ambivalent (meat and sweets; [25,45]) indeed differs on potential ambivalence when being assessed via split semantic scales and compare these results with respective ratings on bipolar valence scales. Next, we investigated whether adding potential ambivalence as a predictor to the multinomial logistic regression model significantly improved the correct classification of univalent, neutral, and ambivalent stimuli compared to the baseline model only including a bipolar valence scale as a predictor. A superiority of the model including potential ambivalence would indicate a conceptual difference between univalence, neutrality, and ambivalence which is not captured in bipolar valence ratings. Finally, we explored whether felt ambivalence and arousal differ for the different categories. This would indicate that univalent, neutral, and ambivalent attitudes due not only conceptually differ, but also lead to different downstream consequences.

## Method

### Inside TUNA: Pictures and their characteristics

To ensure the usability of the database, new pictures were photographed, the TUNA database pictures are published with CC BY-NC-SA 4.0. This copyright allows one to share and adapt the materials for non-commercial use; however, original authors must be credited and the adaptations must be shared under the same terms. A total of 796 pictures were generated for the database. Because all pictures of the TUNA database were exclusively photographed for this purpose, the content is restricted to what was feasible to photograph. However, we tried to include various univalent, neutral, and ambivalent attitude objects to heighten the usability of the database. Most pictures depict food including Vegetables ($n = 178$), Fruits ($n = 87$), Meat ($n = 130$), Meat Substitutes ($n = 18$), Fish and Seafood ($n = 27$), Other Animal Products ($n = 37$), Carbs ($n = 43$), Nuts ($n = 20$), and Sweets ($n = 49$); other pictures depicted inanimate objects including Household Items ($n = 118$), Money ($n = 12$), and Vehicles ($n = 6$); lastly pictures depicted animate objects like Flowers ($n = 19$), and Animals ($n = 52$). For all pictures, the background and angle are held constant. In addition, the images were uniformly formatted so that all images are 600 x 450 pixels. Original-size pictures are also available on the project site on the Open Science Framework (https://osf.io/cyauw/). Because image properties can influence the evaluation of pictures [54], image characteristics for the uniformly formatted pictures were analyzed using the script from Blechert et al. [47] and Matlab T2022a (The Mathworks, Inc. Natick, USA). This allows us to report color (proportion of red, green, and blue averaged across non-white pixels), size (proportion of non-white pixels relative to total pixels), intensity (difference between mean luminance of non-white pixels and the white background), within-

object contrast (standard deviation of luminance across non-white pixels of the grayscaled image) and objective complexity (proportion of outline-related pixels) in the validation table.

For all food pictures, protein, carbs, fat, and kcal per 100g are included in the validation table. This allows comparability of the TUNA and the Food-Pics [47] database. Additionally, for most food pictures there is a low-processed version vs. high-processed version of the food as well as an uncooked vs. cooked version. For example, there is a picture of a whole raw chicken (low processing and uncooked) and a fried chicken (low processing and cooked) as well as chicken sautéed strips raw (high processing and uncooked) and chicken sautéed strips cooked (high processing and cooked).

## Procedure and data collection

Before the data was collected, the procedure was approved by the ethics committee of the local university (EK Nr. 08/2020). Data was collected between November 1st, 2020 and March 31st, 2023. In the informed consent form, participants were informed about the topic of the study, procedure, duration, and data processing. Participants provided written consent by selecting "Yes, I agree" in a drop-down menu. When participants did not consent to the study procedure (i.e., selecting "No, I do not agree (and therefore do not participate)"), they were thanked for their interest and informed that they could only participate after giving consent. After giving informed consent, participants started the study. First, demographic variables were collected (gender, age, native language, occupation). Next, participants provided general information about their diet separately indicating how often they eat meat, fish, or other animal products (e.g., milk, cheese, eggs). The answer options were never (1), once a month (2), once per week (3), multiple times a week (4), and daily (5). Following that, participants indicated if they follow a vegetarian or vegan diet (from here on referred to as veg*an). Next, participants indicated if they were on another diet, for example, for weight loss. Subsequently, data on height and weight were collected, as well as when their last meal was (7 or more hours ago, 5–6 hours ago, 3–4 hours ago, 1–2 hours ago, less than 1 hour ago).

Next, the slider that was used to collect the responses of the participants was explained, and participants were able to familiarize themselves with its use with a test statement ("I understand how to use the slider"). The slider was a long gray line and ranged from 0 to 100 and only the endpoints were labeled (see https://osf.io/rb8a2/ for screenshots of the study pages). The labels depended on the questions. Next, a variety of questions was presented to assess familiarity, recognizability, complexity, arousal, palatability, desire to eat, anger, disgust, empathy, valence, positivity, negativity, and felt ambivalence. For each question, the presentation of the pictures was randomized. Before each block of questions, participants read a brief introduction for the next construct. For example, before answering the arousal question, participants read the following instruction: "In the following, you are going to be asked in what way this object triggers an emotional arousal in you. Please move the slider spontaneously to the position that best describes your current assessment."

Consistent with Blechert et al. [47] data on familiarity (yes vs. no) and recognizability (easy vs. difficult) was collected first with a dichotomous response format. All other responses were indicated on the aforementioned slider and presented in a randomized order. To ensure compatibility with other databases, data on complexity (This object is complex, scale: disagree to agree), arousal (When I see this object I experience emotional arousal, scale: disagree to agree), palatability (This food is palatable, scale: disagree to agree), and desire to eat (I would like to eat this food right now if it were in front of me, scale: disagree to agree) was collected. Note that palatability and desire to eat were only presented for food pictures.

In addition to the commonly used measures, we also assessed anger (When I see this object I experience anger, scale: disagree to agree), disgust (When I see this object I experience disgust, scale: disagree to agree), and empathy (When I see this object I experience compassion, scale: disagree to agree) to extend previous databases by including more affective variables. The questions to differentiate between attitudes (e.g., neutrality and ambivalence) were presented in a fixed attitude question block. Valence (Evaluate this object, scale: very negative to very positive) was always followed by positivity (How positive is this object regardless of its negative aspects?, scale: not positive at all to very positive) and negativity (How negative is this object regardless of its positive aspects?, scale: not negative at all to very negative), which in turn were followed by felt ambivalence (To what extent do you experience conflicting thoughts or feelings towards this object?, scale: not at all conflicted to maximally conflicted). These split semantic scales on positivity and negativity were used to calculate a potential ambivalence score based on the SIM-Index [(Positivity + Negativity)/2 - |Positivity–Negativity|], where higher values stand for more ambivalence, lower values for univalence, and values around zero for neutrality [20]. Note that we assessed potential ambivalence first because potential ambivalence is considered to be a prerequisite of felt ambivalence [23]. The attitude question block was randomly presented with the other questions.

On the final page, participants were thanked for their participation and informed that the goal of the study was to evaluate a picture database. The full procedure of the validation study is depicted in Fig 1.

## Samples

We recruited three heterogeneous samples (see Table 1 for an overview of sample statistics).

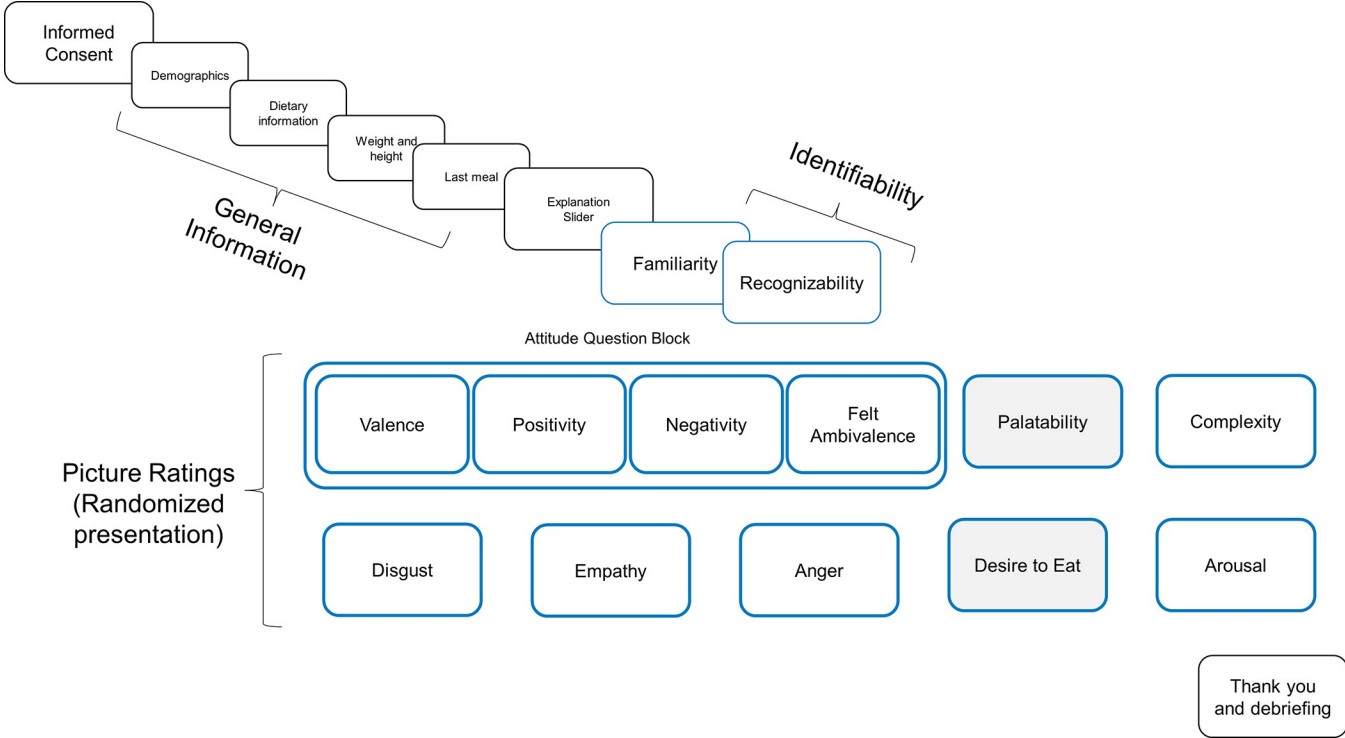

**Fig 1. Depiction of the procedure of the validation studies.** Blue borders indicate the self-reported picture ratings. Within the randomized presentation the sequence of the attitude question block was fixed (i.e., valence was followed by positivity which was followed by negativity which in turn was followed by felt ambivalence). Items with gray backgrounds were only presented for food pictures.

**Table 1. Descriptive statistics for the total sample as well as for the different samples separately.**

|  | Total | Representative US | German Student | Convenience |
|---|---|---|---|---|
| Final $N$ | 3232 | 2058 | 786 | 388 |
| Gender |  |  |  |  |
| Female | 1918 | 1036 | 630 | 252 |
| Male | 1267 | 998 | 154 | 115 |
| Non-binary | 47 | 24 | 2 | 21 |
| $M_{age}$ (SD) | 37.14 (16.72) | 44.37 (16.24) | 22.05 (3.16) | 29.36 (10.49) |
| Eating Style |  |  |  |  |
| Omnivore | 2771 | 1962 | 502 | 307 |
| Veg*an | 461 | 96 | 284 | 81 |
| $M_{BMI}$ (SD) | 25.72 (6.64) | 28.09 (7.41) | 22.45 (3.60) | 24.46 (5.38) |
| Occupation |  |  |  |  |
| School | 39 | 26 |  | 13 |
| College/University | 1129 | 216 | 781 | 132 |
| Apprenticeship | 13 | 8 |  | 5 |
| Employed | 1351 | 1166 | 4 | 181 |
| Pensioner | 230 | 218 |  | 12 |
| Other | 470 | 424 | 1 | 45 |

*Note.* The convenience sample includes the prolific participants who were recruited in a pre-study and the participants recruited via personal communication and social networks. For eating style, vegetarians and vegans were grouped as veg*an, all other eating styles were grouped as omnivores.

1. We collected data from a representative US sample via prolific. This sample consisted of 2,060 participants. Due to incorrect responses to attention check questions two participants were excluded, resulting in a final sample of 2,058 participants.

2. A total of 787 students from Trier University participated in the study in return for course credit. One participant failed the attention check and thus was excluded.

3. Via personal communications as well as via social media (Reddit, Facebook, and websites specifically dedicated to study participation) a convenience sample of 319 participants was recruited. In this sample, we included 69 participants from prolific which were collected to gauge how long the prolific study takes before the data on the representative US sample was collected.

The final sample consisted of 3,232 participants. We provide the full validation ratings and the validation ratings for the representative US and German student samples. Because the convenience sample was rather small, some pictures had zero ratings in this sample. Therefore, validation ratings for the convenience sample are only included in the total sample, the male and female as well as the veg*an and omnivore subsamples. Because participants' demographics might influence the picture evaluation [47], validation ratings are additionally provided split by gender and diet. In sum, we provide validation ratings for the following (sub)samples in the validation tables (see https://osf.io/645jk/ for validation table) and in the TUNA desktop app: total sample, representative US, German student sample, veg*ans, omnivores, females, and males. Sample statistics for these samples are available in S1 Table (https://osf.io/8pj94/).

## Differences in the procedure for the different samples

Data collection differed slightly across the different samples. In the following, we describe the deviations from the procedure, if not stated otherwise, the procedure was the same as

described in the section on procedure and data collection. In the prolific sample as well as the convenience sample, data on the country, political orientation, and ethnicity were additionally collected. Furthermore, instead of only asking if participants were vegetarians or vegans, they self-labeled as a meat eater (I regularly eat meat or fish), meat reducer (I try to rarely eat meat or fish), pescetarian (I do not eat meat but do eat fish), vegetarian (I do not eat meat and fish, but eat other animal products, like eggs or dairy products), and vegan (I do not eat meat, fish or other animal products, like eggs or dairy products), as well as how long they followed their current dietary pattern in years. This variable was recoded so that participants self-labeling as vegetarian and vegan were grouped as veg*an and all others as omnivores.

The picture selection in the German student and convenience sample was completely randomized. In the student sample participants evaluated 10 pictures and in the convenience sample, participants evaluated three pictures (previously the convenience sample evaluated five pictures, however, it took too much time without compensation). Because the secondary aim of the prolific studies was to validate a questionnaire, participants rated six pictures and answered the Meat Ambivalence Questionnaire at the end of the study [51]. To ensure randomization and achieve approximately equal ratings of each picture, the pictures were divided into six groups: meat (130 pictures), vegetables (130 pictures), inanimate objects (136 pictures), fruit and plants (133 pictures), fish, cheese, eggs and vegetables (134 pictures), noodles, baked goods, sweets and animals (133 pictures). Thus, in the prolific survey, participants viewed one randomly selected picture from each group. In the prolific and German student survey, there were also two attention-checks. The attention-checks were two instructed response questions ("This is a check of your attention: Please move the slider completely to the right [left] i.e., "agree" ["disagree"]") and were randomly intermixed with the other question blocks.

Approximately 20 ratings per stimulus are standard for databases [11]. Because some pictures had less than 20 ratings, we collected student data in a follow-up survey which only included these pictures. In this survey, participants evaluated 13 pictures. Thus, the last 36 participants included in the German student sample rated 13 instead of 10 pictures.

## Expectations

By validating the TUNA database, we aimed to provide a standard set of pictures, including various attitude objects that can be distinguished based on more complex attitudes: univalence, neutrality, and ambivalence. To demonstrate the importance of distinguishing complex attitudes, we provide a case study using stimuli that have previously been assumed to be univalent, neutral, or ambivalent. Specifically, it has been argued that fruits are univalent positive [25], carbs like bread or oats are neutral [44] and sweets [25] as well as meat [45] are ambivalent. Our aim was to demonstrate that the validation ratings within the TUNA database align with the anticipated attitudes on split semantic scales, indicating potential ambivalence, but not on bipolar valence ratings. Furthermore, our intention was to illustrate that these variations in attitudes could yield substantial implications if researchers fail to account for them while selecting stimuli.

In our first analysis, we expected that the ambivalent categories (sweets and meat) have higher potential ambivalence than the univalent (fruits) and neutral (carbs) categories. We additionally expected that the neutral category has higher potential ambivalence than the univalent category because negative values on the SIM-Index as an indicator of potential ambivalence suggest univalence, and values around zero suggest neutrality [20]. Hence, we would first replicate that the univalent, neutral, and ambivalent categories used in previous studies are indeed univalent, neutral, and ambivalent [25,44,45].

In addition, we wanted to show that bipolar valence scales might not capture these differences in attitudes. While ambivalent attitude objects typically elicit a rating in the middle of a bipolar valence scale as if they were neutral [20], other ambivalent attitude objects elicit rather positive ratings. For instance, Norris et al. [25] demonstrated that unhealthy food (e.g., sweets) and healthy food (e.g., fruits) have similarly positive ratings on a bipolar valence scale, although, they differ on potential ambivalence when measured via split semantic scales. We, therefore, expected that fruits would not differ from sweets on a bipolar valence scale, and we expected that carbs and meat do not differ on a bipolar valence scale. However, on the same scale, we expect fruits and sweets to have higher valence ratings than carbs or meat. This would demonstrate that bipolar valence scales cannot distinguish between complex attitudes.

Then, we aimed to highlight the added value of potential ambivalence for predicting category membership of the pictures. Using multinomial logistic regression with neutral carbs as reference category, we expect that the prediction for category membership improves when potential ambivalence is added as an additional predictor compared to a baseline model, including only valence measured with a bipolar scale as a predictor. Hence, we expect that the test model, including potential ambivalence as a predictor, explains more variance and has a higher percentage of correctly categorized pictures than the baseline model. Specifically, the superiority of the test model means that taking potential ambivalence into account reduces the confusion between ambivalent and neutral as well as univalent and neutral stimuli. This would reveal that potential ambivalence should be considered when selecting univalent, neutral, and ambivalent stimuli.

Finally, we wanted to demonstrate that univalence, neutrality and ambivalence not only differ in their structure of positive and negative associations but are also associated with different affective responses. Specifically, we expect the ambivalent categories to have higher felt ambivalence and arousal than the neutral and univalent categories [20,23]. The neutral and univalent categories should not differ in their felt ambivalence and arousal. This would highlight that researchers who solely rely on bipolar valence ratings to select stimulus material may overlook significant differences within people's attitudes that can distort the perception of the stimuli.

## Results

All analysis were conducted after the collection of the data and R [55] was used for the data analysis. Each of the 796 pictures was rated at least 20 and maximally 45 times ($M = 27.3$, $SD = 4.84$).

### Identifiability of the pictures

In the present database, we used the dichotomous questions of recognizability and familiarity as indicators of whether people were able to identify the depicted object. Therefore, we coded the response option of not being able to recognize and not being familiar with the object as 0 and being able to recognize and being familiar with the object as 1. Grouping the data by picture—and thus averaging across participants—and multiplying it with 100 led to the percentage of a picture being recognized or being familiar. We found a median of 86.96 for recognizability indicating that half of the pictures had 86.96% or more ratings as easy to recognize and a median of 95 for familiarity indicating that half of the pictures had 95% or more ratings indicating familiarity with the depicted object.

### Differentiating univalence, neutrality, and ambivalence

With the validation data, we first aimed to replicate the univalent, neutral, and ambivalent groups used in previous research and test that these do indeed correspond to the assumed

**Table 2. Means and standard deviations for the self-report scales by category.**

| Category | Neutral Carbs | | Univalent Positive Fruits | | Ambivalent Meat | | Ambivalent Sweets | |
|---|---|---|---|---|---|---|---|---|
| | *M* | *SD* | *M* | *SD* | *M* | *SD* | *M* | *SD* |
| Potential Ambivalence | 0.37 | 9.35 | -11.93 | 14.91 | 6.48 | 9.64 | 9.99 | 7.90 |
| Valence | 66.18 | 10.86 | 71.42 | 16.51 | 39.19 | 9.49 | 66.33 | 9.17 |
| Positivity | 73.11 | 9.38 | 77.52 | 15.38 | 48.74 | 10.50 | 73.61 | 7.38 |
| Negativity | 31.01 | 8.44 | 25.61 | 15.49 | 59.52 | 7.55 | 40.01 | 6.63 |
| Felt Ambivalence | 24.55 | 7.92 | 19.42 | 11.78 | 43.02 | 5.97 | 33.79 | 6.97 |
| Arousal | 23.38 | 8.11 | 26.01 | 7.17 | 40.31 | 8.60 | 37.01 | 7.95 |
| Recognizability | 75.13 | 26.65 | 78.65 | 27.96 | 58.42 | 26.05 | 82.31 | 24.12 |
| Familiarity | 85.61 | 17.11 | 86.34 | 22.13 | 74.94 | 20.27 | 88.19 | 17.09 |

attitude. Previous researchers have used carbs (e.g., oats or bread) as neutral category [44], fruits as univalent positive [25], and meat [45] as well as sweets [25] as ambivalent categories. The pictures in the TUNA database encompass these categories. However, because the number of ambivalent meat (*n* = 130) and univalent positive fruit pictures (*n* = 87) is substantially higher than the number of ambivalent sweets (*n* = 49) and neutral carb pictures (*n* = 43) we randomly sampled 50 ambivalent meat and 50 univalent positive pictures to use in the analyses. This should prevent faulty results due to unequal group size. The random sampling was performed three times and results converged. Thus, we only report the first randomly drawn sample and the analyses for the additional samples can be found in the supplemental material (https://osf.io/8pj94/). The lowest mean for recognizability was 58.42, indicating that, on average, pictures were categorized more often as easy than difficult to recognize, and the lowest mean for familiarity was 74.94, indicating that, on average, the pictures were categorized more often as familiar than unfamiliar. Hence, the average identifiability of the categories was sufficient for the analysis. For all analyses, pairwise comparisons were Bonferroni corrected and means and standard deviations can be found in Table 2.

We first demonstrate that the picture groups differ in their potential ambivalence. Therefore, we calculated a one-factor (category: univalent positive fruit vs. neutral carbs vs. ambivalent meat vs. ambivalent sweets) ANOVA with the SIM-Index as dependent variable. Higher values indicate ambivalence, values around zero neutrality, and negative values univalence [20]. We would expect positive values on the SIM-Index for the ambivalent categories that do not significantly differ, as both should be ambivalent. However, we would expect the ambivalent categories to have higher values on the SIM-Index than neutral carb and univalent positive fruit category. Additionally, we would expect neutral carb pictures to have values around zero which should be significantly higher than univalent positive fruit pictures which should have negative values. Results indicate significant differences in potential ambivalence between the categories, $F(3, 188) = 39.19$, $p < .001$, $\eta^2 = .39$. Consistent with the hypothesis that ambivalent meat and ambivalent sweets have high but similar potential ambivalence, we found positive means for both categories that did not significantly differ ($p = .65$). As expected, these categories had higher potential ambivalence than neutral carb and univalent fruit pictures ($ps < .05$). Additionally, neutral carb had higher potential ambivalence than univalent positive fruit ($p < .001$). For neutral carb pictures, the mean also indicated rather neutral attitudes and for univalent positive fruit pictures the potential ambivalence was negative. Thus, the picture classification of fruits as univalent, carbs as neutral, and meat as well as sweets as ambivalent seems appropriate.

Next, we want to demonstrate that these categories are not distinguishable when using bipolar valence scales. The one-factor (category: univalent positive fruit vs. neutral carbs vs.

ambivalent meat vs. ambivalent sweets) ANOVA with valence as dependent variable indicates that the categories differ in their valence, $F(3, 188) = 74.28$, $p < .001$, $\eta^2 = .54$. Replicating the results of Norris et al. [25] we did not find significant differences between univalent positive fruits and ambivalent sweets ($p = .21$) indicating rather positive valence for both categories. Additionally, ambivalent meat had significantly lower valence than ambivalent sweets ($p < .001$), neutral carb ($p < .001$), and univalent positive fruit pictures ($p < .001$; all other $ps > .22$). Thus, results indicate that a bipolar valence scale cannot differentiate between univalent, neutral, and ambivalent pictures. Additionally, the finding that mere valence is insufficient to differentiate univalence, neutrality, and ambivalence is consistent even when the German student and representative US sample are analyzed separately. This highlights that when selecting stimulus material based on a bipolar valence scale, researchers might inadvertently select ambivalent material instead of univalent or neutral material.

Because the lower valence for the ambivalent meat category was somewhat unexpected, we wondered how the structure of positive and negative associations of ambivalent meat differed compared to the neutral carb, the univalent positive fruit, and the ambivalent sweets category. To explore this, two one-factor (category: univalent positive fruit vs. neutral carbs vs. ambivalent meat vs. ambivalent sweets) ANOVAs with positivity and negativity as dependent variables were calculated. Results indicate that the categories differ in their positivity ($F(3, 188) = 69.23$, $p < .001$, $\eta^2 = .53$) as well as their negativity, $F(3, 188) = 104.57$, $p < .001$, $\eta^2 = .63$. For positivity, only ambivalent meat was significantly less positive than neutral carbs ($p < .001$), univalent positive fruits ($p < .001$) as well as ambivalent sweets ($p < .001$; all other $ps > .35$). For negativity, ambivalent meat also significantly differed from all other categories ($ps < .001$) indicating that meat dishes had the highest negativity. However, ambivalent sweets also had higher negativity compared to neutral carbs ($p < .001$) and univalent positive fruits ($p < .001$). Neutral carbs and univalent positive fruits only differed marginally ($p = .072$) with univalent positive fruits having the lowest negativity. In sum, these results indicate that the positivity scale replicated the bipolar valence scale whereas the negativity scale allows a more nuanced differentiation of attitudes.

## Added value of potential ambivalence for correct attitude classification

Going beyond this replication of previous findings and the exploration of positive and negative associations, we aimed to demonstrate why it is important to go beyond bipolar conceptualizations of valence when selecting stimulus materials. Therefore, we aim to demonstrate that bipolar valence scales are insufficient to predict univalent, neutral, and ambivalent attitudes. Specifically, we use multinomial logistic regression to predict membership to the same categories used above: two typically ambivalent categories in meat and sweets, one univalent positive category in fruits as well as one neutral category in carbs as the reference group. We expected that including potential ambivalence significantly improves the prediction and correct classification of the pictures. Specifically, multinomial logistic regression allows us to model the probabilities of the correct classification for each category. For the baseline model, we only use valence as predictor. We compare this baseline model to a model with valence and potential ambivalence as predictors. Thereby, it is possible to determine the additionally explained variance by adding potential ambivalence.

The first model including only valence as a predictor was significant ($\chi^2 (6) = 118.57$, $p < .001$, AIC = 410.19, *Nagelkerke's $R^2$* = .53) indicating a relationship of 53% between valence and attitude category. Results indicate that higher valence led to a higher probability of a picture being univalent positive fruit compared to neutral carb, however, the lower bound of the 95% confidential interval (CI) of the odds ratio for valence is 1.001 indicating a small effect

**Table 3. Coefficient and odds ratio of multinomial logistic regression for the baseline and the test model.**

|  | | 95% CI for Odds Ratio | | |
| --- | --- | --- | --- | --- |
|  | *B(SE)* | Lower | Odds Ratio | Upper |
| Model 1 | | | | |
| Neutral vs. Ambivalent Meat | | | | |
| Intercept | 9.34(1.89)*** | 505.67 | 11354.11 | 254942.8 |
| Valence | -0.18(0.03)*** | 0.78 | 0.84 | 0.89 |
| Neutral vs. Ambivalent Sweets | | | | |
| Intercept | 0.07(1.11) | 0.12 | 1.07 | 9.47 |
| Valence | 0.0009(0.02) | 0.97 | 1.0009 | 1.03 |
| Neutral vs. Univalent Positive | | | | |
| Intercept | -2.35(1.24)+ | 0.01 | 0.1 | 1.09 |
| Valence | 0.04(0.02)* | 1.001 | 1.04 | 1.07 |
| Model 2 | | | | |
| Neutral vs. Ambivalent Meat | | | | |
| Intercept | 10(1.81)*** | 637.62 | 22121.18 | 767462.5 |
| Valence | -0.19(0.03)*** | 0.77 | 0.82 | 0.89 |
| Potential Ambivalence | 0.02(0.04) | 0.95 | 1.02 | 1.09 |
| Neutral vs. Ambivalent Sweets | | | | |
| Intercept | -5.95(1.99)** | 0.00 | 0.002 | 0.13 |
| Valence | 0.08(0.03)** | 1.03 | 1.08 | 1.15 |
| Potential Ambivalence | 0.13(0.03)*** | 1.07 | 1.14 | 1.21 |
| Neutral vs. Univalent Positive | | | | |
| Intercept | 2.68 (1.59)+ | 0.65 | 14.58 | 326.24 |
| Valence | -0.04(0.02)+ | 0.91 | 0.95 | 1.001 |
| Potential Ambivalence | -0.12(0.03)*** | 0.84 | 0.89 | 0.94 |

*Note.* +$p < .1$.

* $p < .05$.

**$p < .01$.

*** $p < .001$.

(see Table 3). Higher valence led to a higher probability of a picture being neutral carb compared to ambivalent meat. However, valence was not a significant predictor for the ambivalent sweets category. In the baseline model, neutral carbs were never classified correctly, 90% of the ambivalent meat pictures, 46.94% of the ambivalent sweets pictures, and 68% of the univalent positive pictures were classified correctly.

The second model included potential ambivalence as an additional predictor along with valence. This model was also significant ($\chi^2$ (9) = 195.57, $p < .001$, AIC = 328.9, *Nagelkerke's R$^2$* = .73) indicating 20% more explained variance than the baseline model. Furthermore, the model with potential ambivalence was significantly better than the baseline model, $\chi^2$ (3) = 87.29, $p < .001$. In this model, potential ambivalence significantly predicted whether a picture belonged to ambivalent sweets or neutral carbs and whether a picture belonged to the univalent positive fruit or to the neutral carb category (see Table 3). Consistent with previous findings that ambivalent sweets pictures are rather ambivalent, higher potential ambivalence led to a higher probability of a picture being of the ambivalent sweets category compared to neutral carbs. Furthermore, the probability of being classified as neutral carbs was higher when potential ambivalence was higher compared to univalent positive fruits. However, potential ambivalence did not significantly predict whether a picture was neutral carbs or ambivalent meat. In

this model, 34.88% of the neutral carb pictures, 84% of ambivalent meat pictures, 73.35% of the ambivalent sweets pictures, and 74% of the univalent positive fruit pictures were classified correctly. Thus, except for the ambivalent meat category the percentage of correctly classified pictures increased when adding potential ambivalence as a predictor—especially for neutral carb pictures. In sum, the results of the multinomial logistic regression highlight the superiority of the model with potential ambivalence as predictor.

## Univalence, neutrality, and ambivalence elicit different responses

To show the consequences of inadvertently confusing ambivalence with neutrality or univalence, we explore differences between these categories in the experience of conflict as well as arousal. In fact, based on literature [20,23], we expected ambivalent categories to elicit higher felt ambivalence and arousal compared to the univalent positive fruit and the neutral carbs categories. To investigate this, we calculated two one-factor (category: univalent positive fruit vs. neutral carbs vs. ambivalent meat vs. ambivalent sweets) ANOVAs with felt ambivalence and arousal as dependent variables.

Higher potential ambivalence should translate into higher felt ambivalence, thus, we would expect that ambivalent meat and ambivalent sweets have higher felt ambivalence than neutral carbs and univalent positive fruits. Indeed, the results indicated differences between the categories, $F(3, 188) = 74.35$, $p < .001$, $\eta^2 = .54$. Consistent with the hypothesis, results indicated higher felt ambivalence for ambivalent meat and ambivalent sweets compared to neutral carbs and univalent positive fruits, however, ambivalent meat elicited also higher felt ambivalence than ambivalent sweets. Additionally, neutral carbs also elicited higher felt ambivalence than univalent positive fruits (all $p$s $< .02$).

The results of Schneider et al. [20] indicate that ambivalence should elicit higher arousal, therefore, we expected higher arousal for ambivalent meat and ambivalent sweets compared to univalent positive fruit or neutral carbs. Analysis indicates significant differences between the categories, $F(3, 188) = 50.8$, $p < .001$, $\eta^2 = .45$. Results do not indicate significant differences between neutral carbs and univalent positive fruit pictures ($p = .68$) or ambivalent meat and ambivalent sweets ($p = .24$), however, ambivalent meat and ambivalent sweets elicited significantly higher arousal than univalent positive fruits and neutral carbs ($p$s $< .001$). In sum, these results indicate that ambivalent pictures elicit higher experience of conflict (i.e., felt ambivalence) and higher arousal compared to univalent positive fruit and neutral carb pictures.

## Analyses for the representative US and German student sample

We also performed all analyses for the representative US and German student sample separately. Hence, we calculated one-factor (category: univalent positive fruit vs. neutral carbs vs. ambivalent meat vs. ambivalent sweets) ANOVAs with potential ambivalence, valence, positivity, negativity, felt ambivalence, and arousal as dependent variables for the representative US and the German student sample. Consistent with the results for the total sample, we found that univalent positive fruits had the lowest potential ambivalence, followed by neutral carbs. The ambivalent categories had the highest potential ambivalence. Additionally, and as expected, the bipolar valence ratings were insufficient to distinguish univalence, neutrality, and ambivalence. As for the consequences of ambivalence, the results show that the ambivalent categories elicit higher felt ambivalence and higher arousal than the univalent and neutral category. Descriptively, all effects are in the expected direction, however, due to lower measurement accuracy and power in the subsamples, not all Bonferroni corrected pairwise comparisons reached significance. However, the results consistently show that a bipolar valence scale cannot differentiate between univalence, neutrality, and ambivalence. Furthermore, the results show

that these different attitudes lead to different downstream consequences, which can affect how people respond to the pictures.

We also performed multinomial logistic regression analyses for both subsamples separately. The results show the superiority of the test model, including valence and potential ambivalence as predictors. Additionally, adding potential ambivalence as a predictor improves the proportion of correctly categorized pictures (see supplemental material for details https://osf.io/8pj94/). In sum, the separate analyses for the representative US and German student sample replicate the results of the total sample.

## Discussion

Existing databases do not allow a differentiation between univalence, neutrality, and ambivalence. Previous research has demonstrated differences in ambivalence even though valence was rated similarly on bipolar scales [20,25]. Because univalence, neutrality, and ambivalence have different affective, behavioral, and cognitive consequences [16], researchers may thus inadvertently introduce bias in their research results when selecting stimuli based on bipolar valence validation ratings.

To help researchers select appropriate stimuli for their studies, we introduce the TUNA database which incorporates split semantic scales to assess potential ambivalence. Thus, the TUNA database represents the first database to allow a distinction between univalence, neutrality, and ambivalence. The TUNA database consists of typical univalent, neutral, and ambivalent stimuli with a focus on food. Consistent with previous research, our results indicate that typical ambivalent categories such as meat and sweets [25,45] elicit more potential and felt ambivalence compared to rather univalent categories such as fruits [25] or rather neutral categories such as bread or oats. Additionally, multinomial logistic regression analysis indicated that potential ambivalence significantly improved the percentage of correct categorization to these univalent positive, neutral, and ambivalent categories. Surprisingly, however, valence differentiated ambivalent meat from the other categories better than potential ambivalence.

This result is somewhat unexpected because we would expect neutral as well as ambivalent pictures to have valence ratings in the middle of the bipolar valence scale. Exploring positivity and negativity separately, we found that the positivity scale replicated the results of the bipolar valence scale with only the ambivalent meat category having significantly lower positivity compared to the other categories. However, the negativity ratings provided a more nuanced picture. Ambivalent meat had significantly higher negativity than ambivalent sweets which in turn had significantly higher negativity than neutral carbs or univalent positive fruits. Univalent positive fruits and neutral carbs only marginally differ in their negativity with neutral carbs having slightly more negativity. The shift for neutral carb pictures in the positive direction might be explained by the positivity offset. The positivity offset is the tendency to evaluate objects slightly positive when there is no information about the object (i.e., neutral attitude; [17]). The shift in the positive direction of unhealthy foods (i.e., sweets) replicates the results of Norris et al. [25] and the shift in the negative direction for ambivalent meat replicates results from Buttlar et al. [51]. These results indicate that even when a bipolar valence scale indicates rather univalent attitudes the underlying attitude structure might be more complex.

Whereas this indicates that bipolar valence scales are insufficient to differentiate complex attitudes, the data also indicate that confusing these attitudes can lead to downstream consequences for the unambiguous interpretation of the study results. In fact, both ambivalent categories lead to higher experience of conflict and arousal compared to the univalent or neutral category. This corroborates previous theorizing and empirical findings [20,23] and highlights the consequences of mistaking ambivalent with neutral or univalent attitudes (cf. [25]). In fact,

the experience of conflict is thought to trigger a variety of coping processes [16]. The model of ambivalence-induced discomfort [23], for example, proposes that felt ambivalence and the accompanying discomforting arousal may lead to procrastination or biased information processing. Even for simple categorization tasks (e.g., categorizing an object as positive vs. negative), there are response differences between ambivalence and neutrality. That is, responses to ambivalent stimuli are significantly faster than responses to neutral stimuli [24]. Thus, selecting supposedly neutral stimulus material based on bipolar valence ratings might lead to unwanted effects when inadvertently ambivalent material was selected.

The TUNA database enables researchers to select picture stimuli based on validation ratings that allow a nuanced differentiation of attitudes. Results of our analysis indicate that this nuanced consideration of attitude components is necessary to unambiguously interpret research results. However, the TUNA database is not only the first database with validation ratings that allow the differentiation of complex attitudes but also takes a multitude of other factors into account that can influence the evaluation and perception of stimuli. For example, the TUNA database extends previous databases by adding more affective measures including anger, disgust, and empathy. To maximize the utility of the TUNA database, we collected data on variables previously captured in databases [47], for example, valence, arousal, palatability, desire to eat, and complexity. Because the TUNA database also collected ratings on these variables it is compatible with existing databases. Furthermore, the validation table also includes data on physical image characteristics (e.g., intensity or color) and nutritional information for food pictures. This wide range of information about the stimuli allows a diverse use of the TUNA database. From ambivalence and food perception research to investigating disgust—the TUNA database is applicable in many research areas.

## How to access TUNA

Even though the validation tables and pictures are accessible via the supplemental material, we recommend accessing the TUNA database via its accompanying desktop app. This tool is designed to facilitate stimulus selection while simultaneously accounting for the complexity of attitudes, compiling all validation data and images into a unified location. In the first tab, a description of the database and desktop app is presented. In the sidebar, the filter options are located. All filters are preset so that all validation ratings for the total sample are displayed. However, changing a filter option leads to a real-time adaptation of the validation tables. All self-report scales can be used to filter the pictures. These filter options make it easy to exclude pictures, for example, due to lower recognizability. The picture selection resulting from the currently set filters and the associated validation table can be downloaded via the buttons in the bottom left corner (i.e., Download Data or Download Images). This facilitates comparison of picture selection based on different subsamples. Besides the data table with all variables (Full Data Table), the data table with only the self-reported ratings (Self-Reported Variables) or with only the picture characteristics (Image Characteristics) can be selected via the tab menu. In the table, a thumbnail of the picture is displayed beside the validation ratings (see Fig 2).

The code for the underlying shiny app is adapted from the code of the website accompanying the Restrain Food Database [56]. However, users only need to download the folder containing the app and double-click the run.bat. Because R-portable and all R-packages necessary to run the code are installed within the app as well as Firefox-portable users do not need to install any program on their device and the app runs independently from any further updates to R or any changes in the packages (see https://osf.io/fys36/ for a detailed description on how to run the app).

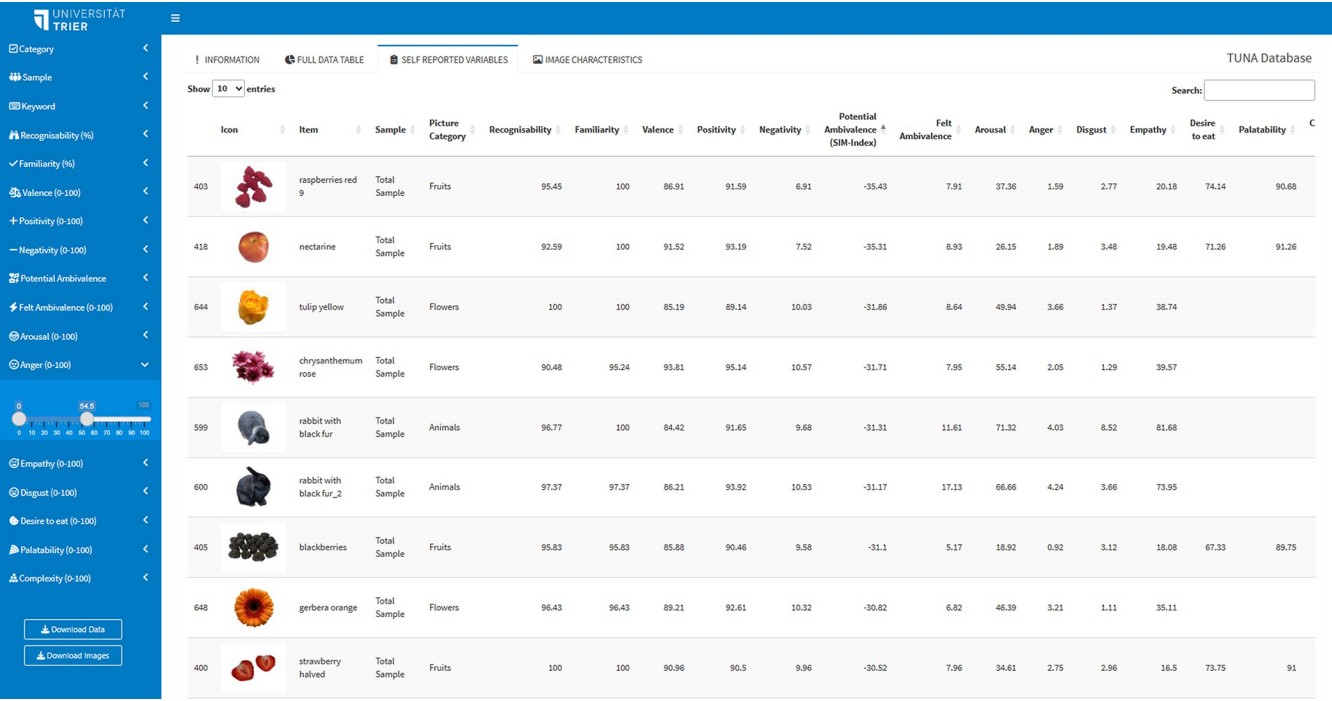

**Fig 2. Screenshot of the TUNA app.** In the sidebar on the left side are the filter options displayed. In this example, all pictures with an anger rating between 0 and 54.5 were selected. At the top of the page the tabs to change to the information side, the full data table, the self-reported variables tables, and the image characteristics are located.

## Limitations

A limitation of the data might be the time of data collection of the student sample. We started data collection in November 2020 and ended in March 2023. This was necessary in order to generate a sufficiently large sample because it was only possible to recruit approximately 300 student participants from the local university per year. Due to this long survey period, it is possible that contextual factors (e.g., change in inflation) have varied and changed the ratings. However, we argue that these influences might also differ when the database is used and thus our ratings might be even better because they are represented in the validation ratings.

Another limitation of the TUNA database pertains to its stimuli, primarily comprising food images. This bias stems from the ease of depicting food and its clear categorization into univalent, neutral, and ambivalent conditions. Using these stimuli, we demonstrated the necessity to differentiate univalence, neutrality, and ambivalence with food stimuli. Even for non-food stimuli incorporated in our database, such as money, vehicles, household items, or flowers, we would expect the same results. Validation ratings measured via split semantic scales, but not bipolar scales should enable researchers to differentiate between neutrality and ambivalence. While researchers might not select a wide variety of non-food stimuli from the database as picture content was limited based on feasibility, the chosen approach also resulted in adopting a CC BY-NC-SA 4.0 copyright. Hence, researchers can still select various univalent, ambivalent, and neutral pictures from the TUNA database which can be shared and adapted for non-commercial use when the original authors are credited, and the adaptations are shared under the same terms.

A third possible limitation of the TUNA database might be the samples used for validation. Even though the separate analyses for the German student and representative US sample

replicate the results of the total sample and highlight that a bipolar valence scale is insufficient to differentiate complex attitudes, there are minor (descriptive) differences in evaluation extremity (see https://osf.io/8pj94/). For example, whereas the ambivalent meat category has higher potential ambivalence than the ambivalent sweets category in the representative US sample, the ambivalent sweets category has higher potential ambivalence than the ambivalent meat category in the German student sample. The divergence between samples might indicate some sample specific aspects that influence the evaluation of the pictures. Such differences might be the higher proportion of veg*ans in the German student sample compared to the representative US sample or the higher BMI in the representative US sample compared to the German student sample. Because Germany and the USA are Western, Educated, Industrialized, Rich, and Democratic (WEIRD; [57]) and the overall effects of our analyses are the same, however, we argue that, it is possible to use the combined ratings. However, the cross-cultural generalization is limited, and further research is needed to confirm the database's validity in other samples.

## Conclusion

The TUNA database advances established picture databases by providing validation ratings that help to differentiate between univalence, neutrality, and ambivalence. The analyses show that bipolar valence scales are insufficient to differentiate univalence, neutrality, and ambivalence, but also that these attitudes lead to different downstream consequences. For example, ambivalent pictures lead to higher experience of conflict and arousal compared to neutral or univalent pictures. The diverse sample used in the validation of the TUNA database allows for versatile use of the database, and the accompanying TUNA desktop app facilitates easy selection of pictorial stimuli. As these validation ratings were conducted on a broad set of attitude objects, we hope that we thereby offer researchers valuable validation data for selecting appropriate stimuli for their studies.

## Supporting information

**S1 Table. Descriptive Statistics for the total sample as well as the subsamples.**
(PDF)

## Acknowledgments

We would like to extend our gratitude to all the friends and family members that waited and ate partly cold food for this project. Additionally, we would like to thank the shops that allowed us to photograph their food, flowers, and animals. Furthermore, we would like to thank Johannes Drobny, David Muhr, Luisa Sabel, and Alina Zumkley for their help in taking and preparing the pictures as well as recruiting the convenience sample. We would also like to thank Tarini Singh for her help in translating the stimulus description and her valuable feedback on earlier versions of the manuscript. Additionally, we thank Bernhard Baltes-Götz for his help in programming the study. Finally, we would like to thank Christopher D. Chambers, Ines Duarte, Mark Randle, Leah Maizey, Loukia Tzavella, and Rachel Condé Adams for providing us with the code of their shiny app which we adapted for the TUNA desktop App.

## Author Contributions

**Conceptualization:** Lena Hahn, Benjamin Buttlar, Eva Walther.

**Data curation:** Lena Hahn, Benjamin Buttlar, Ria Künne.

**Formal analysis:** Lena Hahn.

**Funding acquisition:** Lena Hahn, Benjamin Buttlar.

**Investigation:** Lena Hahn, Ria Künne, Eva Walther.

**Methodology:** Lena Hahn, Benjamin Buttlar, Ria Künne.

**Resources:** Lena Hahn, Benjamin Buttlar, Eva Walther.

**Software:** Lena Hahn.

**Supervision:** Benjamin Buttlar, Eva Walther.

**Validation:** Lena Hahn, Benjamin Buttlar, Ria Künne, Eva Walther.

**Visualization:** Lena Hahn, Benjamin Buttlar, Eva Walther.

**Writing – original draft:** Lena Hahn, Benjamin Buttlar, Ria Künne, Eva Walther.

**Writing – review & editing:** Lena Hahn, Benjamin Buttlar, Eva Walther.

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
