## [Decision Letter · Decision Letter 0]

14 Sep 2023

PONE-D-23-24241Introducing the Trier Univalence Neutrality Ambivalence (TUNA) Database: A picture database differentiating complex attitudesPLOS ONE

Dear Dr. Hahn,

Thank you for submitting your manuscript to PLOS ONE. After careful consideration, we feel that it has merit but does not fully meet PLOS ONE’s publication criteria as it currently stands. Therefore, we invite you to submit a revised version of the manuscript (Major revision) that addresses the points raised during the review process. Please submit your revised manuscript by Oct 29 2023 11:59PM. If you will need more time than this to complete your revisions, please reply to this message or contact the journal office at plosone@plos.org. Please include the following items when submitting your revised manuscript:A rebuttal letter that responds to each point raised by the academic editor and reviewer(s). You should upload this letter as a separate file labeled 'Response to Reviewers'.A marked-up copy of your manuscript that highlights changes made to the original version. You should upload this as a separate file labeled 'Revised Manuscript with Track Changes'.An unmarked version of your revised paper without tracked changes. You should upload this as a separate file labeled 'Manuscript'.

We look forward to receiving your revised manuscript.

Kind regards,

Junchen Shang

Academic Editor

PLOS ONE

Journal Requirements:

Reviewers' comments:

Reviewer's Responses to Questions

**Comments to the Author**

1. Is the manuscript technically sound, and do the data support the conclusions?

Reviewer #1: Partly

Reviewer #2: Yes

2. Has the statistical analysis been performed appropriately and rigorously? 

Reviewer #1: No

Reviewer #2: Yes

3. Have the authors made all data underlying the findings in their manuscript fully available?

Reviewer #1: Yes

Reviewer #2: Yes

4. Is the manuscript presented in an intelligible fashion and written in standard English?

Reviewer #1: Yes

Reviewer #2: Yes

5. Review Comments to the Author

Reviewer #1: In this study the authors introduce a new picture set and a rating method that enables researchers to differentiate between univalence (positive, negative), neutrality and ambivalence.

The main advantages of the study are (1) the high number of participants; (2) international data collection (US vs. German sample); (3) it provides solution to a methodological problem of previous rating scales.

Although the paper has several strengths, I do not recommend it for acceptance in its current form. Please, find some critics below.

ABSTRACT

Affective responses (anger, disgust, empathy) are mentioned in the abstract, but their role and relevance remain unclear in the manuscript. Why did the authors choose these three?

INTRODUCTION

Page 3, lines 60-61: One of the main focus points of the manuscript is the ambivalence – neutrality difference. If possible, please add some more empirical evidence on the need of differentiation (to demonstrate the gap that the current work fills in).

P 6, ls119-127: It is unclear, if the intention was to develop a food-specific dataset? Why were non-food pictures included? What is the reason behind to pick pictures from these categories? Are food pictures and non-food pictures relate to eachother (e.g. through associations, context, cultural aspects etc)?

I’m not sure if all the information in paragraph #2 (p 6, from line 128) should be here. The number of validation ratings, participants and self-report/picture are the results of the validation process, while information in ls 135-138 are demonstarting the goals of the study. Please, modify it or find its place at another part of the manuscript as a conclusion.

P7, l 140: Please, clarify the reason why these three emotions were selected!

METHODS

P 8, from l 169: The number of pictures in each category are unequal, which is not a problem in itself, but it is unclear what was the conceptual background of selection? Why vegetable pictures are so dominant? What is the ratio of food and non-food pictures? (p 15, from l 328 might be the answer...)

P 9, ls 187-188: Please, explain the relevance of adding protein, carb etc. Information of food-pictures? (If there is another conceptual reason of the study that makes it relevant to include these nutrition details, why was it necessary to add non-food pictures to the database?)

P 10, ls 228-232: Without a clear theoretical background in the Introduction, it is hard to see why anger, disgust and empathy ratings are collected. Please, clarify it.

If I understood it correctly, the convenience sample consisted of some US and some German particiapnts. As the US and German sample do have some significant differences (see Table 1.) it might be the case, that this third group shows the mean of the first two samples. From the results it seems to me that the US and German samples have different eating styles, preferences, BMI etc. It looks like a culture-specific aspect.

P 14, from line 297: the procedure is clear for the German sample. Was it the same in the US sample, as well?

P 14, l 302: I suggest to leave out the Meat Ambivalent Questionnaire and the other study. It makes difficult to follow the logic of the current manuscript.

9 14, l 308: What was the attention-check question exactly?

RESULTS

P 15, l 315-316: Please, clarify this sentence.

If the distinction of food and non-food pictures is important, please provide the identifiability according to the sub-categories in the dataset.

P15, l 328-332: Please, clarify the replication aim with previous research more detailed.

P 16, from l 346: It would be more elegant to include predictions/expectations/hypotheses before the results section.

p 17, ls 374-375: this results are really important! In my opinion, this is key point of the manuscript.

Results in Table 2 and Table 3 were calculated from the total sample? I wonder if the results are the same in the US and German samples, respectively? (Based on the differences in Table 1, they should be.)

DISCUSSION

Please, consider to keep the focus of the manuscript only on food material and sub-categories such as meet, fruits, bread, oats etc. (in accordance with p 22, l 478 ‘focus on food’).

Maybe the TUNA App should be the topic of a different publication.

P 25, l 555: How is the ‘cultural background’ defined? Why are the same?

In sum, I do not suggest the manuscript for acceptance in its current form due to some conceptual and methodological concerns. Please, consider (1) if you wish to keep the focus on food pictures only and (2) to investigate the effects in the US and German samples separately.

Reviewer #2: This is a normative study about the affective (an other) properties of a large set of pictures. Its greatest contribution is to assess affectivity, not using a bipolar scale, but unipolar scales of positivity and negativity. The analyses performed demonstrate that using bipolar scales is not the proper procedure for distinguishing neutrality from ambivalence. This has important implications for research on affective processing (which commonly relies on normative datasets), evidencing the need to be very careful in the selection of "neutral" stimuli. Therefore, the paper is a good and useful contribution to the field and deserves to be published. I have, however, some comments that the authors need to consider before the paper is accepted. I list them below:

Introduction

-The normative study is mostly focused on food pictures. The authors need to provide a justification for this choice (which may limit the applicability of the dataset).

-The normative study includes ratings of anger, disgust and empathy. There is no justification for the inclusion of these variables/emotions and not others (e.g., fear, sadness, happiness). The authors should explain the relevance of these variables.

Methods and analyses

-As far as I understand, each participant evaluated a small set of pictures in several variables. The order of the variables was random, except for valence, positivity, negativity and felt ambivalence, which were rated always in the same order. The authors should justify this procedure/order. The assessment of felt ambivalence may have been affected by the previous assessment of the other variables.

-Was a definition of "emotional arousal" provided to participants? If not, it must have been difficult for them to rate this variable. This may have produced a large variability in the ratings.

-Although the dataset is focused on food-related images, it also includes other types of pictures. There are probably univalent, ambivalent and neutral images among them. The authors should perform the same analyses that they have done with food-related images, with these other non-food images. This would inform the readers about the generalizability of the obtained results.

-Why were ratings of disgust, anger and empathy collected if no analysis was performed on these data?

-Why were the participants classified into vegetarian and non-vegetarian if no analysis was performed on these data? Probably, the vegetarian/non-vegetarian condition has affected the ambivalence/univalence ratings (especially for meat). This information should be added to the results section.

6. PLOS authors have the option to publish the peer review history of their article (what does this mean?). If published, this will include your full peer review and any attached files.

Reviewer #1: No

Reviewer #2: No

---

## [Author Response · Author response to Decision Letter 0]

22 Dec 2023

All the reviewer points are addressed extensively in the ‘Response to Reviewers’ letter our reply in italics and blue. We tried to respond to the remarks in as much detail as possible and provided quotes within our response to the reviewers to outline the changes in the manuscript. Within the manuscript changes are highlighted via the track changes mode in MS Word the manuscript. If anything is unsatisfactory, however, we would be happy to provide further revisions.

---

## [Decision Letter · Decision Letter 1]

16 Apr 2024

Introducing the Trier Univalence Neutrality Ambivalence (TUNA) Database: A picture database differentiating complex attitudes

PONE-D-23-24241R1

Dear Dr. Hahn,

We’re pleased to inform you that your manuscript has been judged scientifically suitable for publication and will be formally accepted for publication once it meets all outstanding technical requirements.

Kind regards,

Junchen Shang

Academic Editor

PLOS ONE

Additional Editor Comments (optional):

Reviewers' comments:

Reviewer's Responses to Questions

**Comments to the Author**

1. If the authors have adequately addressed your comments raised in a previous round of review and you feel that this manuscript is now acceptable for publication, you may indicate that here to bypass the “Comments to the Author” section, enter your conflict of interest statement in the “Confidential to Editor” section, and submit your "Accept" recommendation.

Reviewer #1: All comments have been addressed

Reviewer #2: (No Response)

2. Is the manuscript technically sound, and do the data support the conclusions?

Reviewer #1: Yes

Reviewer #2: (No Response)

3. Has the statistical analysis been performed appropriately and rigorously? 

Reviewer #1: Yes

Reviewer #2: (No Response)

4. Have the authors made all data underlying the findings in their manuscript fully available?

Reviewer #1: Yes

Reviewer #2: (No Response)

5. Is the manuscript presented in an intelligible fashion and written in standard English?

Reviewer #1: Yes

Reviewer #2: (No Response)

6. Review Comments to the Author

Reviewer #1: (No Response)

Reviewer #2: (No Response)

7. PLOS authors have the option to publish the peer review history of their article (what does this mean?). If published, this will include your full peer review and any attached files.

Reviewer #1: **Yes: **Anita Deak

Reviewer #2: No

---

## [Editor Report · Acceptance letter]

29 Apr 2024

PONE-D-23-24241R1 

PLOS ONE

Dear Dr. Hahn, 

I'm pleased to inform you that your manuscript has been deemed suitable for publication in PLOS ONE. Congratulations! Your manuscript is now being handed over to our production team.

Kind regards, 

on behalf of

Dr. Junchen Shang 

Academic Editor

PLOS ONE